# Structural and functional characterization of the Sin Nombre virus L protein

**Kristina Meier[1], Sigurdur R. Thorkelsson[2,3]☉, Quentin Durieux Trouilleton[4]☉, Dominik Vogel[1], Dingquan Yu[2,5], Jan Kosinski[2,5,6], Stephen Cusack[7], Hélène Malet[4,8], Kay Grünewald[2,3,9], Emmanuelle R. J. Quemin[2,3]¤, Maria Rosenthal**[1,2,10]*

**1** Bernhard Nocht Institute for Tropical Medicine (BNITM), Hamburg, Germany, **2** Centre for Structural Systems Biology (CSSB), Hamburg, Germany, **3** Leibniz Institute of Virology (LIV), Hamburg, Germany, **4** University Grenoble Alpes, CNRS, CEA, IBS, Grenoble, France, **5** European Molecular Biology Laboratory (EMBL), Hamburg, Germany, **6** Structural and Computational Biology Unit, European Molecular Biology Laboratory (EMBL), Heidelberg, Germany, **7** European Molecular Biology Laboratory (EMBL), Grenoble, France, **8** Institut Universitaire de France (IUF), Paris, France, **9** University of Hamburg, Department of Chemistry, Hamburg, Germany, **10** Fraunhofer Institute for Translational Medicine and Pharmacology (ITMP), Discovery Research ScreeningPort, Hamburg, Germany

☉ These authors contributed equally to this work.
¤ Current address: Department of Virology, Institute for Integrative Biology of the Cell (I2BC), Centre National de la Recherche Scientifique (CNRS) UMR9198, Gif-sur-Yvette, France
* rosenthal@bnitm.de

**Data Availability Statement:** Coordinates and structure factors or map included in this paper have been deposited in the Worldwide Protein Data Bank (wwPDB) and the Electron Microscopy Data

## Abstract

The *Bunyavirales* order is a large and diverse group of segmented negative-strand RNA viruses. Several virus families within this order contain important human pathogens, including Sin Nombre virus (SNV) of the *Hantaviridae*. Despite the high epidemic potential of bunyaviruses, specific medical countermeasures such as vaccines or antivirals are missing. The multifunctional ~250 kDa L protein of hantaviruses, amongst other functional domains, harbors the RNA-dependent RNA polymerase (RdRp) and an endonuclease and catalyzes transcription as well as replication of the viral RNA genome, making it a promising therapeutic target. The development of inhibitors targeting these key processes requires a profound understanding of the catalytic mechanisms. Here, we established expression and purification protocols of the full-length SNV L protein bearing the endonuclease mutation K124A. We applied different biochemical *in vitro* assays to provide an extensive characterization of the different enzymatic functions as well as the capacity of the hantavirus L protein to interact with the viral RNA. By using single-particle cryo-EM, we obtained a 3D model including the L protein core region containing the RdRp, in complex with the 5′ promoter RNA. This first high-resolution model of a New World hantavirus L protein shows striking similarity to related bunyavirus L proteins. The interaction of the L protein with the 5′ RNA observed in the structural model confirms our hypothesis of protein-RNA binding based on our biochemical data. Taken together, this study provides an excellent basis for future structural and functional studies on the hantavirus L protein and for the development of antiviral compounds.

Bank (EMDB) with the following accession codes: PDB ID 8CI5, EMD-16670. The AlphaFold2 model of the CBD was deposited in ModelArchive with the ID ma-l64m5 (https://www.modelarchive.org/doi/10.5452/ma-l64m5).

**Funding:** We acknowledge funding for this collaborative project by the Leibniz Association's Leibniz competition programme (grant K72/2017). D.Y. and J.K. were supported by the DFG grant KO 5979/2-1. In the framework of this project, S.R.T. benefited from a travel grant from the Leibniz Institute of Virology; H.M. is supported by an endowment of the Institut Universitaire de France; E.R.J.Q. was supported by an individual fellowship from the Alexander von Humboldt Foundation and a Klaus Tschira Boost Fund; M.R. received funding from the German Federal Ministry for Education and Research (grant 01KI2019). Part of this work was performed at the Cryo-EM multi-user Facility at CSSB, headed by K.G. and supported by the UHH and DFG (grants INST 152/772-1, 774-1, 775-1 and 776-1). The funders had no role in study design, data collection and analysis, decision to publish, or preparation of the manuscript.

**Competing interests:** The authors have declared that no competing interests exist.

## Author summary

Hantaviruses are globally distributed zoonotic viruses and can cause severe disease in humans. New World hantaviruses, including Sin Nombre virus (SNV), are endemic to the Americas and cause cardiopulmonary syndromes with fatality rates of up to 35%. There are currently no effective vaccines or specific FDA-approved treatments available. A promising drug target would be the viral multifunctional large (L) protein, which plays a key role in viral genome transcription and replication. We succeeded in expressing and purifying the full-length L protein of SNV and present the first high-resolution structural model of the SNV L protein core region, including the RNA-dependent RNA polymerase, in complex with viral RNA. Additionally, we developed different biochemical assays to investigate protein-RNA interaction, RNA polymerase activity and endonuclease function of the full-length L protein. Based on these assays, we developed a model for viral promoter binding, which is supported by our structural data. The developed protocols and assays for the structural and functional characterization of the hantavirus L protein will facilitate further mechanistic studies and support the development of antiviral strategies.

## Introduction

Sin Nombre virus (SNV) belongs to the *Hantaviridae* family of the order *Bunyavirales*, a large and diverse order of segmented negative-strand RNA viruses (sNSVs) containing several emerging pathogens [1]. The global distribution and severity of human disease as well as the high economic burden inflicted by bunyaviruses overall highlight the need for specific and efficient therapeutic countermeasures, which are to this day not available, to control bunyavirus outbreaks. Additionally, research opportunities are limited, as many hantaviruses must be handled under strict biosafety precautions. SNV is the most prevalent human-pathogenic hantavirus in North America and, like other New World hantaviruses, causes hantavirus cardiopulmonary syndrome with fatality rates of up to 35% [2–4].

The hantavirus genome is tri-segmented and encodes only four structural proteins: the nucleoprotein (N), encoded on the small genomic RNA segment; the glycoprotein precursor, which is co-translationally cleaved into the two glycoproteins Gn and Gc and is encoded on the medium genomic segment; and the L protein, encoded on the large genomic segment (L segment) [5].

The key player in hantavirus replication is the large multifunctional L protein which, together with the nucleoprotein N and viral RNA, forms viral ribonucleoprotein complexes (RNPs). RNPs are the structural and functional units of viral genome replication and transcription in the host cell cytoplasm [6]. The L protein harbors the RNA-dependent RNA polymerase (RdRp), an endonuclease, and likely contains a cap-binding domain (CBD), analogous to the L protein of other bunyavirus families and the polymerase complex of influenza viruses [7]. The endonuclease and cap-binding functions are required for viral transcription initiation via cap-snatching to provide short, capped RNA primers derived from host cell mRNAs [8]. Genome replication is thought to be initiated *de novo* employing a prime-and-realign mechanism and proceeds via a positive-strand antigenomic RNA intermediate [9,10].

The coding regions of each segment are flanked by untranslated regions (UTRs) which contain the RNA promoter and are highly complementary, allowing the 3′ and 5′ termini of the genomic segments to interact via Watson-Crick-Franklin base pairing [11].

Its key roles in the virus life cycle make the L protein a promising target for therapeutic intervention. However, a profound knowledge of the structure, functions and their regulation

are essential for the targeted search for antivirals inhibiting this multifunctional protein. Furthermore, identifying structural and functional commonalities between the hantavirus L protein and L proteins of other bunyavirus families may even permit the development of broad-spectrum antivirals. However, due to their complexity and high molecular weight of 250–450 kDa, full-length bunyavirus L proteins remain difficult to study, and structural data is still limited to the representatives of three bunyavirus families: the *Arenaviridae*, *Phenuiviridae*, and *Peribunyaviridae* (recently reviewed in [7]). In the case of the *Hantaviridae*, these difficulties are further aggravated by the highly active endonuclease domain which suppresses protein expression [12].

In this study, we aimed to characterize the L protein of New World hantavirus SNV structurally and functionally. We established an expression and purification protocol, allowing us to obtain highly stable SNV L protein with the endonuclease active site mutation K124A from insect cells. Our *in vitro* biochemical analysis showed that (i) the SNV L protein specifically binds to both 3′ and 5′ RNA termini, (ii) the RdRp is active in the presence of divalent metal ions and the 3′ end of the RNA promoter as a template, and (iii) the endonuclease retains residual activity in the presence of divalent metal ions despite the active site mutation K124A. Notably, the 5′ RNA terminus was bound with a higher affinity than the 3′ RNA alone, which could be recruited by the presence of the 5′ RNA. By mutational analysis, we determined nucleotides 13 and 14 of the 5′ RNA end to be essential for this specific recruitment of the 3′ RNA to the L protein. However, the RdRp activity *in vitro* was inhibited by the presence of the 5′ RNA.

Using single-particle electron cryo-microscopy (cryo-EM) we obtained structural data of the L protein in complex with the 5′ promoter RNA. Although the flexible N- and C-terminal regions were not resolved, analysis of AlphaFold2 predictions of the unresolved regions suggest the presence of a structurally conserved CBD in the C-terminus. Taken together, we provide new structural and functional insights into the key protein driving hantavirus genome replication and transcription. These results will serve as a foundation for further studies delineating bunyavirus genome replication and transcription and provide guidance for structure-based drug development.

## Results

The full-length SNV L protein (GenBank accession ALI59818.1) carrying the endonuclease mutation K124A was used for protein expression in insect cells via the baculovirus expression system [13,14]. The protein was purified to homogeneity (see Materials and Methods, S1 Fig) and used for biochemical studies and cryo-EM experiments.

### RNA promoter binding to the SNV L protein

The 3′ and 5′ termini of bunyavirus genomes are conserved within bunyavirus families and highly complementary [7]. It was therefore assumed, that both termini form a panhandle-like structure via Watson-Crick-Franklin base pairs, with which the L protein engages. However, in recent years, distinct 3′ and 5′ RNA binding sites were reported for La Crosse virus (LACV, *Peribunyaviridae*), Machupo virus (MACV, *Arenaviridae*), Lassa virus (LASV, *Arenaviridae*), and severe fever with thrombocytopenia syndrome virus (SFTSV, *Phenuiviridae*) L as well as the influenza virus polymerase complex [15–21].

Using electrophoretic mobility shift assays (EMSAs), we detected interaction of the SNV L protein with both 3′ and 5′ RNA terminal nucleotides 1–18 of the L segment (Fig 1A). The L protein-RNA complexes differed in their migration behavior in the native gel, suggesting that binding of the 3′ and 5′ RNAs, respectively, induce distinct conformations of the L protein. In

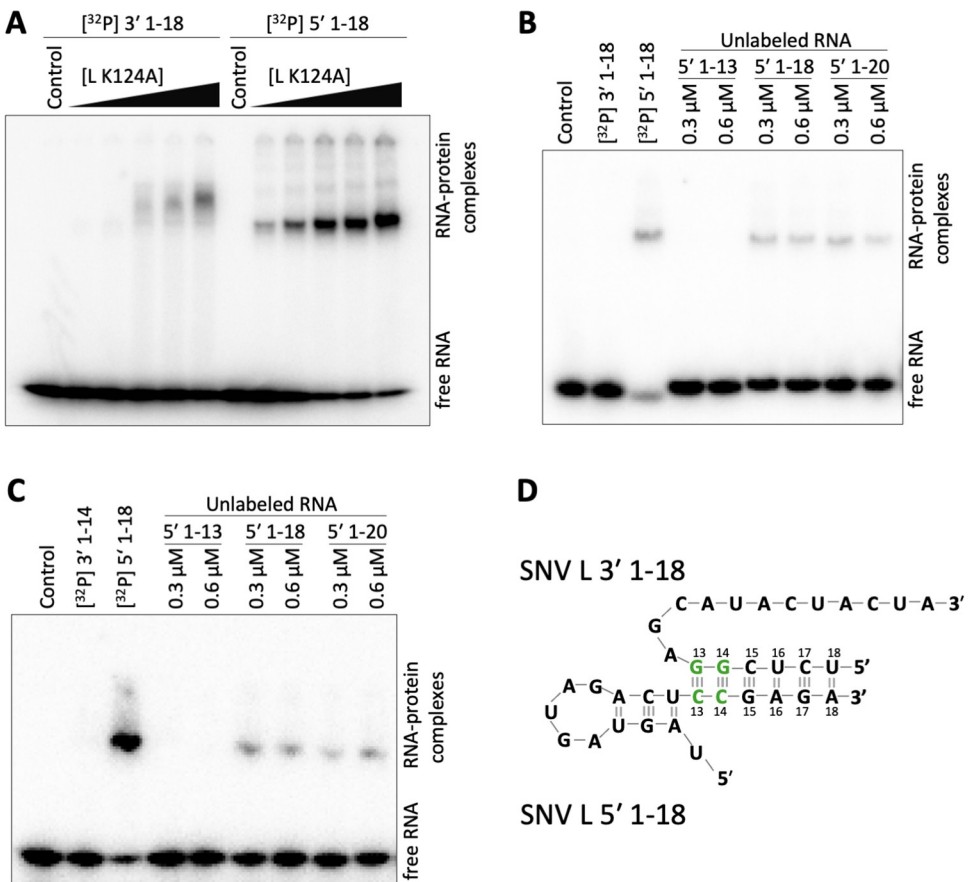

**Fig 1. RNA promoter binding to the SNV L protein. (A)** Electromobility shift assay to determine the binding capacity of the SNV L protein to the 3′ and 5′ promoter RNA ends of the genomic L segment (nucleotides 1–18). Radiolabeled 3′ RNA 1–18 (3′-AUCAUCAUACGAGGCUCU-5′) or 5′ RNA 1–18 (5′-UAGUAGUAGACUCCGAGA-3′) at a concentration of ~0.3 μM was incubated with increasing amounts of L protein (0.2–1.0 μM). The RNA-protein complexes were separated from free RNA by native PAGE and signals were visualized via phosphor screen autoradiography with a Typhon scanner (GE Healthcare). The control lanes show promoter RNA without L protein. **(B, C)** Electromobility shift assay to determine the recruitment of 3′ promoter RNA to the L protein in the presence of 5′ promoter RNA. Labeled **(B)** 3′ 1–18 (3′-AUCAUCAUACGAGGCUCU-5′), or **(C)** 3′ 1–14 (3′-AUCAUCAUACGAGG-5′) promoter RNA at a concentration of ~0.3 μM was incubated with 0.2 μM L protein in the presence of unlabeled 5′ RNA 1–13 (5′-UAGUAGUAGACUC-3′), 1–18 (5′-UAGUAGUAGACUCCGAGA-3′), or 1–20 (5′-UAGUAGUAGACUCCGAGAUA-3′) at an equimolar concentration or a two-fold excess of 5′ to 3′ RNA. Labeled 5′ promoter RNA was used as a positive control, a sample lacking L protein was used as a negative control. **(D)** Schematic depiction of the 5′ promoter RNA forming the characteristic hook-structure and a distal duplex with the 3′ promoter RNA.

contrast to the clearly defined bands formed by the L protein-5′ RNA complex, the SNV L-3′ RNA complex produced more diffuse bands that migrated more slowly in the gel. This indicates a more compact shape of the L protein-5′ RNA complex. In addition, the interaction of SNV L with the 5′ RNA appeared to be stronger than with the 3′ RNA. These findings are consistent with previous data published on SFTSV L [22]. In other bunyaviruses, the terminal 9–10 nucleotides of the 5′ RNA were demonstrated to form a so-called hook-like RNA secondary structure by intramolecular base pairing, that binds to a specific pocket of the L protein. It was, therefore, likely that this feature was also present in hantaviruses.

We used the L protein at a molar concentration that had not resulted in a gel shift of the 3′ RNA in the previous experiment and incubated it with both 5′ and 3′ RNAs. We found that

the labeled 3′ RNA 1–18 was efficiently recruited to the L protein in the presence of unlabeled 5′ promoter RNA 1–18 or 1–20, producing a distinct shifted band as observed with the 5′ RNA alone (Fig 1B). This is in line with structural data on other bunyavirus L proteins and the influenza virus polymerase complex, showing the formation of an RNA duplex between the 3′ and the 5′ nucleotides upstream of the 5′ hook, denoted the distal duplex. This distal duplex serves to direct the 3′ terminus into the RdRp active site for priming [15,16,19,20,23]. As it was unclear how exactly the 3′ and 5′ RNAs were structured and interacting in hantaviruses, we used various shorter RNAs to define the essential elements and interactions of the SNV 3′ and 5′ promoter RNAs. When incubating the shorter 5′ RNA 1–13 with SNV L and the labeled 3′ RNA 1–18, we did not observe any gel shift, although the 5′ RNA 1–13 itself could bind to the L protein (S2A Fig). This indicates that 5′ RNA nucleotides 1–13 were not sufficient for 3′ RNA recruitment. RNA secondary structure prediction using RNAstructure [24] suggested the formation of a 5'-terminal hook-like secondary structure within the terminal 12 nucleotides of the 5' end, with nucleotides 2–4 forming intramolecular base pairs with nucleotides 10–12 (S3 Fig). This would be in line with our findings regarding 3′ RNA recruitment and was subsequently confirmed by structural data (see below). Interestingly, when performing the EMSA with the unlabeled 5′ RNA 1–18 or 1–20 and a shorter, labeled 3′ RNA 1–14 we detected a shift of the shorter 3′ RNA, albeit with less efficiency compared to the recruitment of 3′ RNA 1–18 (Fig 1C). The recruitment was lost when an even shorter 3′ RNA 1–12 was used (S2B Fig). According to the 5′ hook prediction, in the combination 5′ RNA 1–18 and 3′ RNA 1–14 only two base pairs could be formed between the 3′ and 5′ RNA, most likely between nucleotides 13 and 14 of both RNAs. Based on these data, we propose the following model for the SNV promoter RNA: The 5′-terminal 12 nucleotides of the genome form a hook-structure that is bound by the L protein inducing the observed compact protein conformation. The 3′ RNA forms a distal duplex with the 5′ RNA upstream of nucleotide 12, which enhances binding of the 3′ RNA to the L protein with base pairs between nucleotides 13 and 14 being the minimal requirement for 3′ RNA recruitment (Fig 1D).

## Full-length SNV L K124A displays moderate endonuclease activity

The hantavirus L protein endonuclease was previously reported to be highly active and to hamper its own expression, which could be overcome by inserting single mutations into the active site [12,25]. The isolated endonuclease of ANDV L bearing the K124A mutation was shown to still be moderately active but no data has yet been published on the endonuclease activity in the context of the full-length hantavirus L protein. It would be conceivable that the full-length L protein contains specific regulatory mechanisms to control this enzymatic activity as observed for LASV and SFTSV L proteins [19,20].

Using a ribonuclease assay with a radiolabeled PolyA$_{27}$ RNA substrate, we observed a reduction of signal intensity of the substrate band via autoradiography in the presence of increasing concentrations of MnCl$_2$ or MgCl$_2$ but it was difficult to detect distinct degradation products (Fig 2A). This substrate degradation was not observed in presence of the specific endonuclease inhibitor 2,4-Dioxo-4-phenylbutanoic acid (DPBA) [25,26], suggesting that the degradation was indeed catalyzed by the endonuclease domain. Additionally, a fluorescence-based ribonuclease assay was established with various Cy5-labeled RNA substrates, in which we also detected degradation of the Cy5-labeled viral 3′ and 5′ RNAs 1–18 in sub-stochiometric RNA concentrations relative to the L protein (Fig 2B), albeit with lower efficiency than the polyA substrate (S4 Fig). This may be explained by preferred binding of the viral RNA to the L protein to the distinct 3′ and 5′ RNA binding sites or the RdRp active site, resulting in protection of the bound RNA from degradation by the endonuclease. In summary, we could

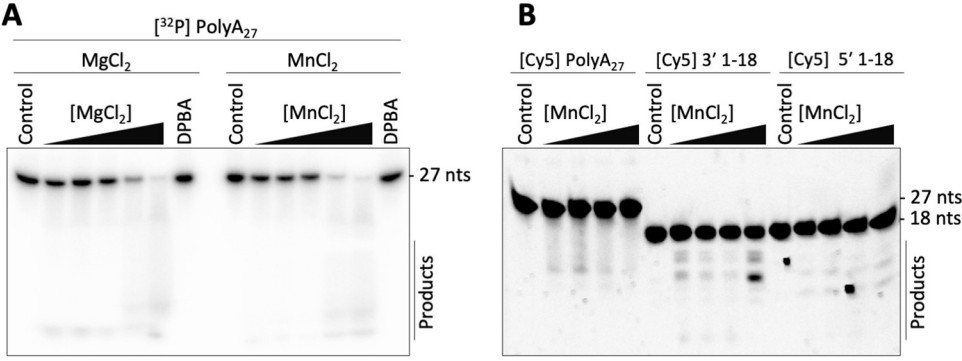

**Fig 2. *In vitro* endonuclease activity of SNV L K124A. (A)** 0.5 μM full-length SNV L K124A were incubated with ~0.3 μM radiolabeled PolyA$_{27}$ RNA substrate in the presence of increasing concentrations of the indicated divalent metal ions (2.5–12.5 mM) and incubated at 37˚C for one hour. Reaction products were separated by denaturing PAGE and signals were visualized via phosphor screen autoradiography with a Typhon scanner (GE Healthcare). The control lanes show substrate RNA without L protein. As an additional negative control, 200 μM DPBA were added to a sample containing 12.5 mM of the indicated metal ions. **(B)** The ribonuclease assay was repeated with 1.5 nM fluorescently labeled RNA, 0.25 μM L protein and varying concentrations (0.5, 1.0, 5.0, or 10.0 mM) of MnCl$_2$.

show that even in the context of the full-length L protein, the SNV endonuclease appears to be active *in vitro* despite a mutation in its active site (*i.e.*, K124A). This assay setup is particularly suited for the testing and development of allosteric endonuclease inhibitors and would also allow studies on the regulatory mechanisms of the endonuclease activity in context of the full-length L protein as observed for other L proteins [19,20].

## The SNV L protein contains an active polymerase

To characterize the enzymatic RdRp activity of the SNV L protein, we carried out *in vitro* RNA synthesis assays with radiolabeled [α]$^{32}$P-GTP for the detection of RNA products by autoradiography as previously described [22,27]. We used the first 3′-terminal 18 nucleotides of the L segment RNA as a template with a 3′-phosphate to avoid elongation of the input RNA. SNV L RdRp was active in the presence of different concentrations of MgCl$_2$ and highly active at MnCl$_2$ concentrations up to 5 mM. MnCl$_2$ concentrations higher than 5 mM resulted in a loss of product signal intensity (Fig 3). Since the endonuclease was found to be active at these concentrations of divalent metal ions (Fig 2), potentially causing degradation of the products, the known endonuclease inhibitor DPBA which efficiently inhibits the hantavirus endonuclease (Fig 2A) [25,26] was added to a sample containing the highest metal concentration tested in this assay. However, the addition of DPBA did not increase the product signal intensity, suggesting that the loss of observed RdRp activity was not caused by product degradation by the endonuclease domain.

Overall, we detected a variety of RNA products generated by the SNV L RdRp although only a single template RNA of 18 nucleotides was used. In the samples with the highest polymerase activities, we observed two main product bands of approximately 20 and 40 nucleotides. Since the elongation of the input RNA was unlikely due to the addition of a 3'-terminal phosphate, the larger band may correspond to the product of two replication rounds, resulting in a product twice the length of the template RNA. Interestingly, we also observed two distinct bands around 20 nts in size, which differed in their intensity depending on whether manganese or magnesium ions were present in the polymerase reaction. This observation might relate to differences in the prime-and-realign mechanism or other enzymatic activities of the L protein depending on the ions present.

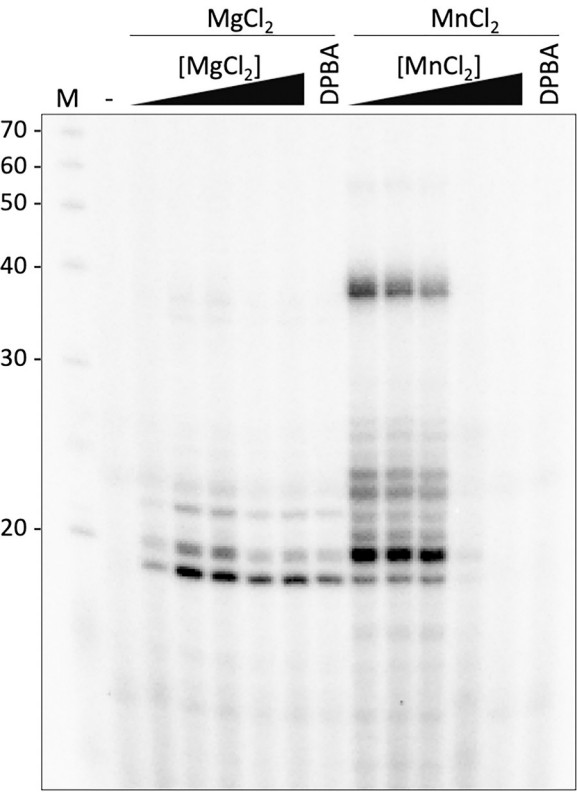

**Fig 3.** *In vitro* **RdRp activity of SNV L K124A.** Full-length SNV L K124A was incubated with SNV L 3′ RNA 1-18P (3′-P-AUCAUCAUACGAGGCUCU-5′) as a template and NTPs supplemented with $[\alpha]^{32}$P-GTP in the presence of increasing concentrations of the indicated divalent metal ions (1.0, 2.5, 5.0, 7.5, or 10 mM), with 10 mM of the indicated metal ions in the presence of 200 μM DPBA, or without divalent metal ions (-) at 30°C for one hour. Reaction products were separated by denaturing PAGE and signals were visualized by phosphor screen autoradiography.

As in the context of RNPs, both promoter ends are expected to be bound by the L protein, we next tested the influence of the 5′ promoter RNA on the RdRp activity of SNV L. Interestingly, the addition of the 5′ RNA 1-18P led to a significant reduction of RNA synthesis in a dose-dependent manner (Fig 4). This was unexpected as for other bunyavirus L proteins a stimulatory role of the 5′ RNA had been described [22,27–29]. We suspected that a very stable distal duplex formed between 3′ and 5′ RNA disturbed RNA production, as previously shown for LACV L [17]. However, a similar effect was observed when 5′ RNA 1–13 comprising only the RNA hook, was used and preincubated with the protein prior to the addition of the template RNA. Therefore, this inhibitory effect is likely not caused by interaction of the 3′ and 5′ RNA forming a distal duplex but potentially by the interaction of the 5′ RNA and the protein itself, as an allosteric inhibitor (Fig 4). This effect is particularly puzzling as the expected product of the 3′ 1–18 template would constitute a 5′ 1–18 complementary RNA which would be expected to form a similar 5′ hook structure and therefore would be a very similar allosteric inhibitor. Alternatively, the observed inhibitory effect of the 5′ RNA 1–13 may be caused by 3′-5′ duplex formation within nucleotides 1–13. However, the 5′ 1–18 complementary RNA product, once released, would also be expected to form a perfect duplex with the 3′ template, which should theoretically have the same effect. Such a feedback inhibition would be difficult to dissect in our assay setup and we cannot be entirely sure that the product is indeed released from the RdRp active site. We conclude that this unexpected effect is likely an artifact of the *in vitro* assay.

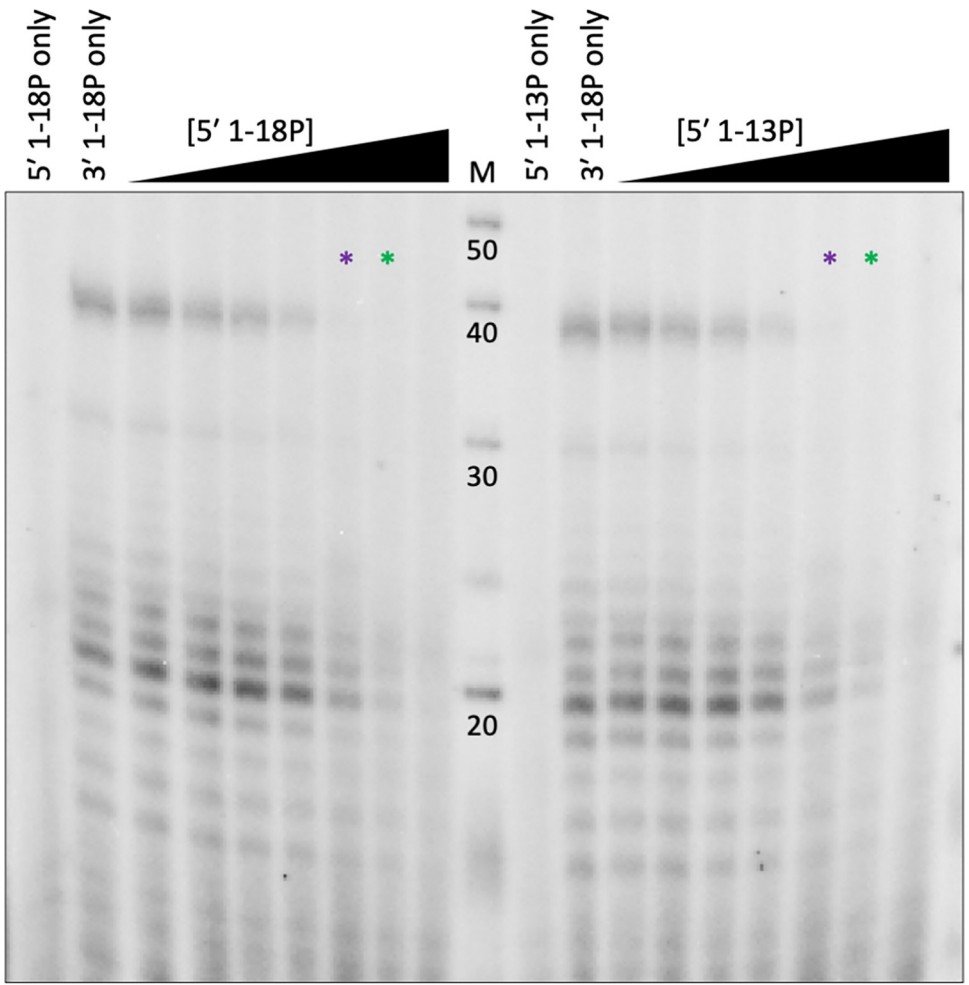

**Fig 4. Influence of the 5′ promoter RNA on SNV L polymerase activity.** Full-length SNV L K124A (4 pmol) was incubated with increasing concentrations (0.5, 1.0, 2.0, 3.0, 4.0, 6.0, or 9.0 pmol) of SNV L segment 5′ RNA 1-18P (5′-UAGUAGUAGACUCCGAGA-P-3′), or 5′ 1-13P (5′-UAGUAGUAGACUC-P-3′) prior to the addition of 6 pmol SNV L segment 3′ 1-18P RNA (3′-P-AUCAUCAUACGAGGCUCU-5′) as a template. NTPs supplemented with [α]$^{32}$P-GTP were added, and the reactions were incubated at 30°C for one hour. Reaction products were separated by denaturing PAGE and signals were visualized by phosphor screen autoradiography. The asterisks mark the samples in which the 5′ RNA concentration was equimolar to the protein (purple) or 3′ RNA (green) concentration.

Since the previous assays relied on the *de novo* initiation of RNA synthesis (Figs 3 and 4), we next tested the capacity of the SNV RdRp to elongate short RNA primers. When the dinucleotide primer AG or the trinucleotide primers UAG, AGU, or GUA and 3′-phosphorylated 3′ RNA 1–18 template were present, we detected RNA synthesis. The product band patterns of primed and unprimed reactions were similar, but the bands were shifted in the gel as expected and according to the annealing site of the respective primer on the template (Fig 5A and 5D). This shift indicates that the products were indeed generated by primer extension. Of note, although the template contains more than one binding site for each primer, no smaller products resulting from initiation at the additional binding sites downstream of the 3′ terminus were detected. However, we cannot exclude the possibility that the primers initiated RNA synthesis internally and were subsequently realigned. The product signal intensity was significantly increased in the reaction containing the AGU primer in comparison to the unprimed reaction. Interestingly, the previously observed inhibition of the RdRp activity in the presence

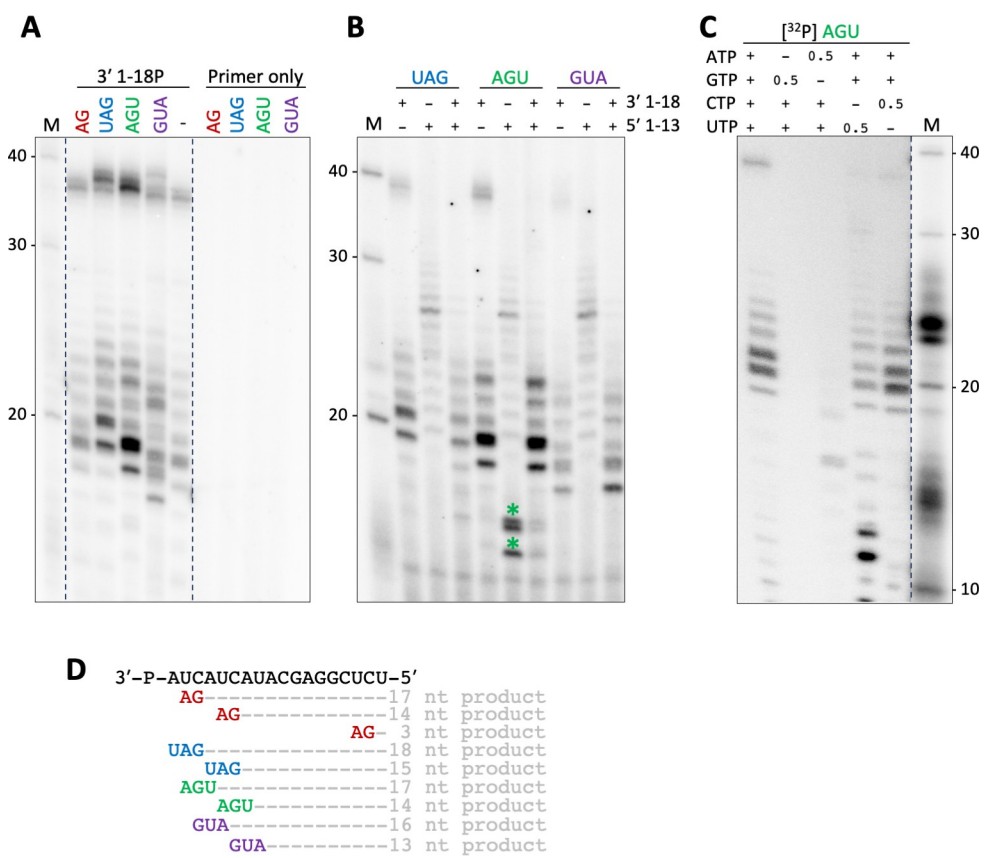

**Fig 5. Primer-elongation RdRp activity of SNV L K124A. (A)** SNV L segment 3′ RNA 1-18P RNA (3′-P-AUCAUCAUACGAGGCUCU-5′) was incubated with a tenfold excess of the indicated primers for 15 minutes prior to the addition of L protein and NTPs supplemented with [α]32P-GTP. Reactions were incubated at 30°C for one hour. Reaction products were separated by denaturing PAGE and signals were visualized via phosphor screen autoradiography. The lane marked with (-) shows an unprimed reaction. **(B)** SNV L was incubated with SNV L segment 5′ RNA 1-13P (5′-UAGUAGUAGACUC-P-3′) as indicated for 15 minutes prior to the addition of equimolar concentrations of SNV L segment 3′ RNA 1-18P and primers as indicated. The assay was carried out as described above. The products marked with asterisks were likely caused by the underlined binding site for the AGU primer on the 5′ RNA 1-13P. **(C)** Primer elongation activity of SNV L RdRp in the absence of different nucleotides. The radioactively labeled AGU primer was used for product detection instead of [α]32P-GTP. The dotted lines indicate cutting of the lanes for presentation purposes. The contrast of the marker lane in **(C)** was adjusted as the intensity of the marker was higher than that of the product bands. **(D)** Sequence of the template RNA and potential primer annealing sites including the resulting product length.

of the 5′ promoter RNA was less apparent in the presence of primers, hinting towards an inhibitory role on *de novo* initiation in the *in vitro* assay setup (Fig 5B).

We next tested the specificity of the SNV L protein RdRp by depriving the RdRp of individual NTPs. To avoid the substitution of the missing NTP with the other corresponding purine or pyrimidine NTP, the concentration of the corresponding NTP was also reduced. We used a radiolabeled [32P] AGU primer for the detection of primer extension products to make this assay independent of individual radiolabeled NTPs. We did not observe primer extension products in the condition lacking ATP and only two faint product bands were detected in the absence of GTP (Fig 5C). These faint bands were shorter than the main products from a reaction containing all NTPs, suggesting that these were abrogation products. The RNA products generated in the absence of UTP or CTP resembled those generated in the reaction containing all nucleotides, suggesting that UTP and CTP can be more readily substituted than other

NTPs. In the absence of CTP, we observed additional shorter product bands which were likely abrogation products. Taken together, our data suggest that the SNV L RdRp has a low specificity for pyrimidines, indicative of a low RdRp fidelity.

## Structure determination of the SNV L protein via cryo-EM

To obtain structural information, we used the recombinantly expressed and purified SNV L protein for the preparation of grids for cryo-EM. In the absence of viral RNA, the L protein adopted a preferred orientation on cryo-EM grids so that the data could not be used for high-quality reconstructions of a model. Imaging of the L protein-RNA complex was complicated by the fact that the L protein-RNA complex was more stable in low salt conditions (S2A Fig) whereas the recombinant L protein itself was most stable in high-salt buffers and precipitated during cryo-EM grid preparation in low-salt conditions. Applying PEGylation [30], we succeeded to improve grid preparation in lower-salt conditions of the L protein in complex with the 5′ RNA 1–18. Single-particle reconstruction yielded a map at 3.2 Å resolution (S6 Fig), comprising parts of the endonuclease linker, the core lobe and vRNA binding lobe (vRBL), the RdRp core with the fingers, palm and thumb domains as well as the bridge, thumb ring and lid domains (Fig 6A). No density was visible for the N- and C-terminal regions containing the endonuclease and, presumably, a CBD, respectively. This suggests that these regions are highly

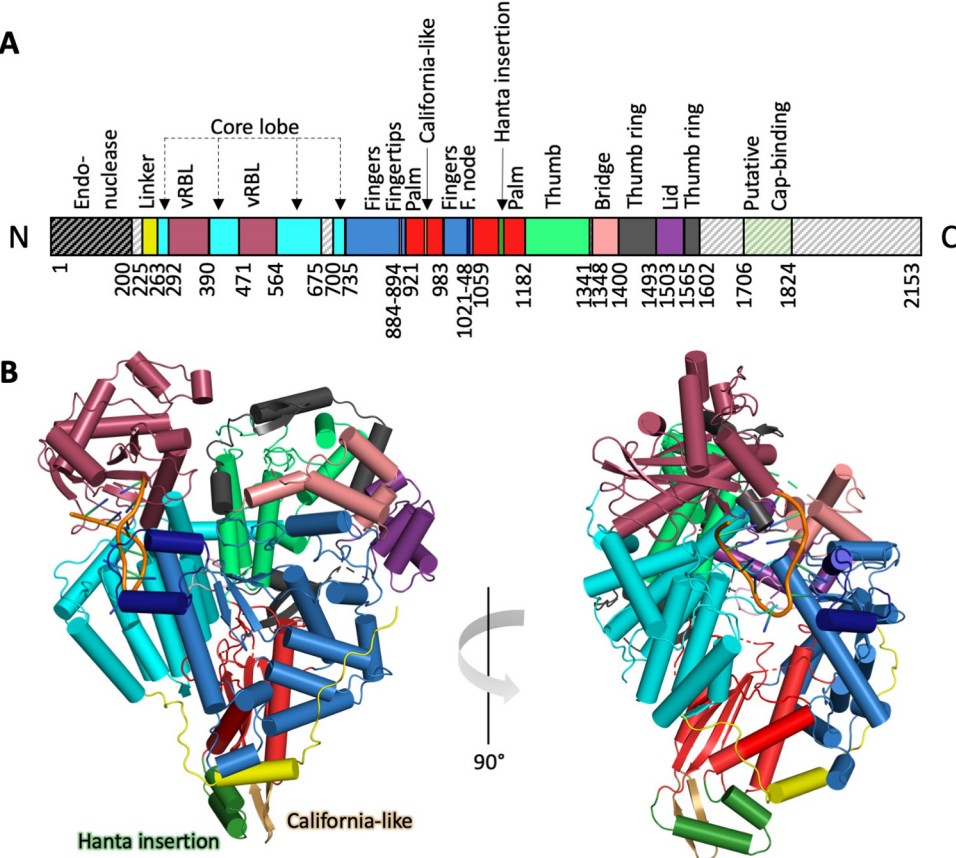

**Fig 6. Structure of the SNV L protein in complex with 5′ RNA. (A)** Linear schematic representation of the domain architecture of the SNV L protein. **(B)** Illustration of two different views of the SNV L structure based on cryo-EM as a cartoon representation. The color code for the different domains is given in **(A)** (PDB 8CI5).

flexible in the 5′ RNA-bound conformation of the L protein, as previously reported for other bunyavirus L proteins [15,16,19,20,22] (reviewed in [7]). As hypothesized from the biochemical studies (see above), nucleotides 2 to 12 of the 5′ RNA were found to be bound in a hook-like conformation to a pocket delineated by the vRBL, core lobe and fingers domains.

## SNV L structure and comparison to related bunyavirus L proteins

SNV L shares structural similarities with the reported L protein structures of related bunyaviruses (Fig 6A and 6B). The similarity to LACV L was especially striking, which is consistent with phylogenetic data as the *Peribunyaviridae* are closely related to the *Hantaviridae*. The endonuclease, previously reported to be located within the N-terminal 200 residues [25,26], was not resolved in the cryo-EM map. The flexible endonuclease linker (residues 225–263) was partially visible in the map wrapping around the L protein core. Compared to the available structures of LACV, LASV, and SFTSV L proteins, the linker of SNV L appears to be more disordered [15–17,19,20,22]. The region spanning from the endonuclease linker to the fingers domain of the RdRp core is composed of a mainly α-helical core lobe (residues 263–291, 401–470, and 564–734 with missing interpretable density for residues 436–448, and 676–699) as well as the vRBL (residues 292–389 and 471–563). The vRBL is structurally most similar to that of LACV and consists of a β-sheet surrounded by several α-helices. In LACV L, the vRBL contains a clamp and an arch that contact the bound 3′ and 5′ RNAs, respectively. The region corresponding to the LACV clamp is smaller in SNV L, spanning only 30 residues (343–372) compared to 45 residues in LACV L. No interpretable density was visible for residues 390–400. This region may correspond to the arch, which is also less well defined in LACV L structures with bound 5′ RNA [15,16].

The core lobe and vRBL are followed by the fingers, palm and thumb domains that are characteristic of sNSV RdRps. The fingers domain spans residues 735–920 and 983–1058 and contains the fingertips (residues 884–894) and finger node (1021–48 with missing density for residues 1032 and 1033) insertions that are conserved among sNSVs and the *Bunyavirales*, respectively. Both insertions were well defined in the map and the loop connecting the two short α-helices of the finger node appears to stabilize the bound 5′ promoter RNA. The α-ribbon insertion in LACV L, which protrudes from fingers domain forming a long and a short α-helix, is absent in SNV L. This region is more similar to the short and disordered α-ribbon-like domain in SFTSV L but is further reduced in SNV L to only 11 residues (835–845).

The palm (residues 921–982 and 1059–1181, missing interpretable density for residues 928–930, 975–977, and 1098–1100) also closely resembles the corresponding domain in LACV L, including a California insertion-like extension [15] (residues 950–963) which forms a β-hairpin in SNV L in contrast to the α-helical hairpin in LACV L. Interestingly, we observed a second insertion (Hanta insertion, residues 1113–1135), protruding from the central β-sheet as two α-helices, which is not present in LACV L or other bunyavirus L proteins of known structure. This insertion appears to be conserved between Old World and New World clades and seems to be specific to hantaviruses (see S1 Alignment).

The thumb consists of an α-helical bundle spanning residues 1182–1340 and is followed by the bridge which, like in LACV L, is formed by four α-helices (residues 1348–1399). The region connecting the thumb to the bridge, in case of LACV L, contains the priming loop. This structure is highly flexible and therefore only ordered in certain conformations. It is possible that a priming loop is also present in the equivalent position in SNV L, however, the SNV map was lacking density corresponding to residues 1319–1331 and 1341–1347. Sequence alignment showed partial conservation among hantaviruses and peribunyaviruses within this region although peribunyavirus L proteins have 11 additional residues that are lacking in hantavirus L proteins (see S1 Alignment).

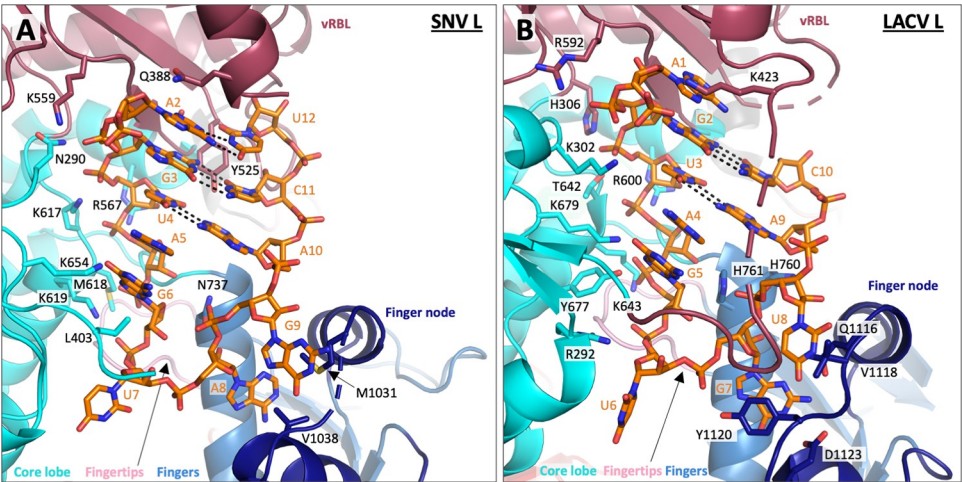

**Fig 7. SNV and LACV L protein 5′ RNA binding sites.** Comparison between 5′ RNA binding to **(A)** SNV L and **(B)** LACV L (PDB 5AMQ). The 5′ RNA binding sites are shown as cartoon representations with domains colored according to Fig 6, and RNA is shown as orange sticks. Amino acid side chains potentially involved in protein-RNA binding are shown as sticks.

The thumb ring, which consists of several α-helices and β-strands, surrounds the thumb domain and is partially visible in the map (residues 1400–1602 with missing density for residues 1459–1482, 1494–1502 and 1568–1569). Inserted into the thumb ring is the lid (residues 1503–1565) which is structurally conserved between SNV L and LACV L and consists of three short parallel α-helices packed against one long α-helix. The C-terminal portion of the protein spanning residues 1603–2153 is completely absent in the map, indicating a high flexibility of this region as observed for other sNSV L proteins [15,19,20,22,28,31].

The viral 5′ RNA is bound in a pocket formed by the vRBL, core-lobe and fingers domains. Nucleotides 2–12 of the 18mer RNA are visible in the map and form a hook-like conformation stabilized by three base pair interactions between A2 and U12, G3 and C11, and U4 and A10 (Figs 7A and S7). A similar hook-like conformation of the 5′ RNA has been found in several bunyavirus L proteins, stabilized by 1–2 intramolecular base pairs. Protein-RNA contacts between the 5′ hook and SNV L include the side chains of Q388, Y525, and K559 from the vRBL, N290, L403, R567, K617, M618, K619, and K654 from the core lobe, N737 from the fingers domain as well as M1031 and V1038 from the finger node (Fig 7A). A similar mode of 5′ RNA binding has been observed in LACV L (Fig 7B).

## 3D structure prediction of the unresolved L protein C-terminus suggests the presence of a cap-binding domain

The N- and C-terminal regions of SNV L K124A were not resolved in the obtained cryo-EM map, likely due to their high degree of flexibility in the chosen conditions. While crystal structures of the isolated N-terminal endonuclease domain have been published for Old World and New World hantavirus L proteins [25,26], no structural information is available for the C-terminal region of the hantavirus L protein. Since hantaviruses, like other sNSVs, rely on cap-snatching to initiate transcription, this region is assumed to harbor a CBD analogous to that within the PB2 subunit of the heterotrimeric influenza virus polymerase complex. Structural evidence of a CBD was recently reported in the C-terminal L protein region of members of the *Arenaviridae*, *Peribunyaviridae*, and *Phenuiviridae* [16,17,19,22,32,33].

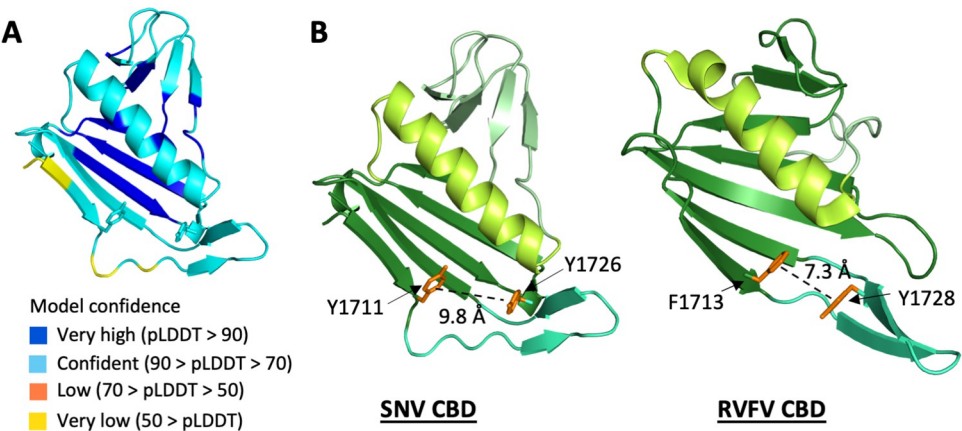

**Fig 8. Putative cap-binding domain predicted within the C terminus of SNV L. (A)** The structure of the C-terminal region of SNV L was predicted using AlphaFold2, visualized in PyMol and searched for a structural motif resembling the published cap-binding domains of bunyaviruses and orthomyxoviruses. The predicted structural model is colored based on the confidence as determined by AlphaFold2's predicted local distance difference test (pLDDT score). A residue with a pLDDT score greater than 90 (blue) indicates high estimated accuracy of the position of its backbone and sidechain rotamers. A pLDDT score above 70 (cyan) suggested that the prediction of the backbone is confident. **(B)** Structural comparison of the putative cap-binding domain within the predicted C-terminal region of the hantavirus SNV L protein and the known structure of the phenuivirus RVFV L cap-binding domain (PDB 6QHG). The black dotted line indicates the distance measurement between the two aromatic amino acid side chains shown as orange sticks, that may be involved in forming the aromatic sandwich to bind the cap.

We predicted the structure of the unresolved C-terminal region of SNV L using AlphaFold2 [34] and found a structural motif characteristic of a CBD, consisting of a β-sheet packed against an α-helix between amino acids 1706 to 1824. This region was mostly predicted with high confidence except for the first β-strand (Fig 8A). Comparison with the reported CBD of Rift Valley fever virus (RVFV, *Phenuiviridae*) L protein revealed high structural similarity (Fig 8B) although the similarity on sequence level is only about 42% (S9 Fig). Similar to RVFV CBD, aromatic sidechains protrude from the tip of the first strand (Y1711) and the adjoining β-hairpin (Y1726), which may be involved in the formation of an aromatic sandwich to stack the cap. The distance between these two aromatic sidechains in the predicted structure was 9.8 Å and therefore larger than in the RVFV CBD (7.3 Å). However, the β-hairpin has a flexible hinge and may be differently positioned relative to the central β-sheet upon ligand binding or stabilization by the remaining L protein regions. Additionally, the first β-strand, which harbors one of the aromatic residues, has a lower pLDDT score in the prediction, indicating that structural details might differ.

## Discussion

The hantavirus L protein is a key player in the replication cycle of this important virus family. We present here the first high resolution partial structure of a New World hantavirus L protein in complex with 5′ RNA along with an extensive biochemical characterization. Binding of the 5′ RNA to the L protein detected in our *in vitro* assays appeared to occur more efficiently than of the 3′ RNA, which is consistent with data published for SFTSV L [22]. In contrast, the arenavirus L protein was found to bind well to both RNAs [27,35].

The ability of SNV L to generate RNA products *de novo* in the presence of the 3′ RNA template only, shows that SNV L does interact with this RNA. However, binding of the 3′ RNA to the RdRp active site in the absence of NTPs may be too unstable to be efficiently visualized by native gel electrophoresis. It should be noted that this assay does not allow for the

differentiation between 3′ RNA binding in the RdRp active site and binding in the putative 3′ RNA secondary binding site.

The 3′ RNA is recruited to the L protein in the presence of the 5′ RNA as detected in EMSAs. However, the addition of the 5′ RNA showed an inhibitory effect on RNA synthesis. This is in strong contrast to published data on LASV L and SFTSV L, which require the presence of the 5′ RNA for *de novo* RNA synthesis [22,27]. For LACV L, it was demonstrated that complementarity of the template 3′ RNA to the 5′ RNA led to less product formation *in vitro* compared to a reaction with an adapted 5′ RNA with less complementarity to the template 3′ RNA. However, in this case, presence of a 5′ RNA was necessary for RdRp activity [17]. The RdRps of Machupo virus L (MACV, *Arenaviridae*) and RVFV L were reported to be active with 3′ RNA only, although the activity was markedly enhanced upon the addition of 5′ RNA [28,29,35]. Thus, while the requirement of the 5′ RNA for RdRp activity varies between the previously studied bunyavirus L proteins, no inhibitory effect of the 5′ RNA compared to a reaction with 3′ template RNA only has yet been reported. This inhibitory effect observed for SNV L RdRp was consistent when 5′ RNA 1–13, comprising the RNA hook only, or 5′ RNA 1–18, containing 5 additional nucleotides forming a distal duplex with the 3′ RNA, were used. Although the distal duplex can be excluded as the cause of this effect, it is conceivable that the high complementarity of the RNA nucleotides 1–13 leads to the sequestration of the template 3′ RNA. Likewise, it is possible that the inhibitory effect is a result of the interaction between the 5′ RNA hook and the L protein. Interestingly, the inhibitory effect was reduced in the presence of short primers, suggesting that only *de novo* initiation of RNA synthesis is inhibited in the presence of the 5′ RNA. Alternatively, primers may compete with the 5′ RNA for binding to the template RNA. However, the highly complementary 5′ RNA would be expected to form a more stable duplex with the template than the short primers at equimolar concentrations as used in our assays. What remains puzzling is that the *in vitro* RNA product would indeed represent a 5′ complementary RNA expected to form a similar hook structure to be bound to the 5′ hook binding site. We would therefore expect an RdRp inhibition by the product, which we are, however, unable to investigate in our assay setting. We speculate that this 5′ inhibitory effect is an artifact of our *in vitro* assay setup, as a similar difficulty in using highly complementary 3′ and 5′ RNAs has been described for LACV L *in vitro* polymerase assays [17].

While a model for the full influenza virus transcription cycle is available [36], the late stages of RNA replication and transcription, including termination, by bunyavirus L proteins remain less well studied. During influenza virus genome transcription, the 5′ RNA remained tightly bound in its dedicated pocket even during termination. This was not the case during influenza virus genome replication as, in contrast to the shorter mRNA, vRNA and cRNA are exact complementary copies of one another so that the 5′ RNA must be released and pass through the RdRp active site. The exact mechanism of 5′ RNA release during genome replication remains to be determined. While some bunyaviruses require the bound 5′ RNA to initiate RNA synthesis *de novo*, the inhibitory effect of the 5′ RNA observed for SNV L on *de novo* initiation, but not primer-dependent RNA synthesis may suggest that the bound 5′ RNA favors initiation of transcription over genome replication in hantaviruses. Clarification of this observation will require structures of the hantavirus L protein in functional states of genome replication and transcription.

Another observation was that the SNV L RdRp more readily substituted pyrimidines than purines when deprived of single nucleotides. The importance of purines is evident from the fact that purines are required to initiate RNA synthesis *de novo*, which was circumvented by the use of a primer in our assay [37]. Our results indicate that purines may be incorporated with a higher precision in primed RNA synthesis as well. This might simply be due to their larger size relative to pyrimidine bases, which may allow for more stabilizing interactions of

the protein with the incoming nucleotide. The inhibitory effect in samples lacking purines, particularly ATP, may be aggravated by the fact that ATP would be the first nucleotide to be added to the short RNA primer in this assay, directly followed by GTP. Since other bunyavirus L proteins were reported to undergo conformational changes between initiation and elongation stages [16,17,19,20], it is possible that the hantavirus L protein incorporates nucleotides with a higher fidelity in the initiation conformation than in the elongation conformation. High fidelity at the initiation stage would be beneficial to ensure the correct sequence of the 5′ terminus, which is an essential regulatory element of genome replication and transcription.

It should be noted that we observed a number of different RNA products rather than one specific product band corresponding to the length of the 3′ template RNA. The use of a 3′-phosphorylated template reduced the number of products observed in our initial assay (S5A Fig) but still resulted in two main product bands, one of which appeared to be twice the length of the other. A similar observation was previously made for the RdRps of polio virus (*Picornaviridae*) and bovine viral diarrhea virus (*Flaviviridae*) generating RNA products that were larger or even multimeric relative to the template length and likely resulted from template switching while the RdRp remained associated with the product [38,39]. Template switching may also occur in our assay as our *in vitro* setup is lacking any termination signals. We also observed a weak product band that was roughly three times the length of the smallest main product band in some assays (S5B Fig). It is, therefore, conceivable that the larger main product band of the SNV L RdRp is the result of template switching, likely an artifact of our *in vitro* assay design. Therefore, the SNV L RdRp appears to be rather unspecific under the conditions of our *in vitro* assay. Our assay setup thus allows for the assessment of the overall activity of the SNV L RdRp but is limited in the identification of the different products.

The structural analysis of the SNV L protein was complicated by the preferred orientation the protein adopted on cryo-EM grids, which could be overcome by PEGylation [30]. This problem has not yet been reported for any other bunyavirus L proteins, at least not to the extent observed for SNV L. The structural model of SNV L presented in this study shares a common overall architecture with other bunyavirus L proteins and is structurally closest to LACV L of the *Peribunyaviridae* [15–17]. Strikingly, we observed a β-hairpin insertion at the same position within the palm domain as the α-helical California insertion found in LACV L. This was unexpected since the California insertion is a feature that is not conserved among the *Peribunyaviridae* but only found in members of the California serogroup and its function remains unknown [15]. In addition to this California insertion-like element, SNV L contains an α-helical insertion that protrudes from the palm domain and is positioned parallel to the California insertion-like β-hairpin. No similar insertion has yet been reported in any other bunyavirus L protein structural study.

The 5′ promoter is bound in a hook-like conformation, similar to LACV, LASV, and SFTSV L proteins as well as the influenza virus polymerase, although the hook is formed by three base pairs in the case of hantaviruses as compared to only one in LASV and SFTSV [19,20], two in LACV [15] or four in influenza virus [21].

Since hantaviruses, like other sNSVs rely on cap-snatching to initiate transcription, SNV L is assumed to harbor a C-terminal CBD, analogous to other bunyavirus L proteins and the influenza virus polymerase complex [8,16,19,22,32,33,40]. Although the C-terminal region was not resolved in our cryo-EM map, a structural prediction of this region carried out with Alpha-Fold2 [34] resulted in a structure that showed striking similarity to known sNSV cap-snatching CBDs. This putative CBD is structurally similar to that of RVFV L with two aromatic side chains protruding from the central β-sheet and an adjacent β-hairpin respectively, to potentially bind to a cap via a stacking interaction [33]. In the case of LACV L, the second aromatic residue is replaced by an arginine residue that protrudes from a disordered loop replacing the

β-hairpin that is present in SNV L and RVFV L [17] (S8 Fig). It seems, therefore, that the CBD structure of hantaviruses is more closely related to phenuiviruses than to peribunyaviruses, which is also supported by comparison on sequence level (S9 Fig). This structural conservation of the RdRp core and the CBD between hantaviruses and peribunyaviruses or phenuiviruses, respectively, is highly interesting from an evolutionary perspective.

In conclusion, we established expression and purification procedures, *in vitro* biochemical assays as well as a protocol for the structural analysis of the multifunctional L protein-RNA complex of SNV. This work constitutes a solid basis for further structural and functional studies investigating the hantavirus mechanisms of genome replication and transcription and future drug development.

# Materials and methods

## Cloning, expression, and purification of full-length SNV L K124A

The L gene of SNV (GenBank accession: KT885044.1) was chemically synthesized (GenScript). The gene was amplified and the point mutation K124A as well as a C-terminal StrepII-tag were inserted by PCR. The gene was cloned into a modified pFastBacHT B vector using the NEBuilder HiFi DNA Assembly kit (New England Biolabs). After transformation of the resulting plasmid into DH10EMBacY *Escherichia coli* cells containing a bacmid as well as a transposase-encoding plasmid, recombinant bacmid was isolated and transfected into Sf21 insect cells for the production of recombinant baculovirus. Protein expression was carried out in Hi5 insect cells. Harvested cells were resuspended in lysis buffer (50 mM HEPES(NaOH) pH 7.0, 500 mM NaCl, 10% (w/w) glycerol, 2 mM dithiothreitol), supplemented with 0.05% (v/v) Tween20 and protease inhibitors (cOmplete, Roche), lysed by sonication, and centrifuged twice at 20,000 x g for 30 min at 4°C. The supernatant containing soluble protein was loaded on Strep-TactinXT beads (IBA) and eluted with 50 mM biotin (AppliChem) in elution buffer (50 mM HEPES pH 7.0, 500 mM NaCl, 10% glycerol, 50 mM Biotin, 2 mM dithiothreitol) or in EM elution buffer (50 mM HEPES pH 8.0, 500 mM NaCl, 10% glycerol, 50 mM Biotin, 2 mM dithiothreitol) for the samples used in cryo-EM. The L protein-containing fractions were pooled and diluted with an equal volume of dilution buffer (20 mM HEPES(NaOH) pH 7.0 or 8.0 for EM samples) before loading onto a heparin column (HiTrap Heparin HP, GE Healthcare). The protein was eluted with heparin elution buffer (50 mM HEPES(NaOH) pH 7.0, 1500 mM NaCl, 10% (w/w) glycerol, 2 mM dithiothreitol) or EM heparin elution buffer (50 mM HEPES(NaOH) pH 8.0, 1000 mM NaCl, 10% (w/w) glycerol, 2 mM dithiothreitol) to reduce the salt concentration for cryo-EM samples. Eluted protein was either directly used for biochemical assays or flash frozen in liquid nitrogen and stored at -80°C. The cryo-EM sample was further purified on an S200 size exclusion chromatography column (GE Healthcare) in GF buffer (30 mM HEPES(NaOH) pH 8.0, 300 mM NaCl, 2 mM dithiothreitol).

## Electrophoretic mobility shift assay

RNAs were chemically synthesized (Biomers) and labeled with $[\gamma]^{32}$P-ATP (Hartman Analytic) using T4 polynucleotide kinase (Thermo Fisher). Labeled RNA was subsequently separated from unincorporated $[\gamma]^{32}$P using Microspin G25 columns (GE Healthcare). RNA was heated to 95°C for 3 min and cooled down on ice to generate single-stranded RNA. Samples containing ~3 pmol labeled RNA and 0–10 pmol L protein were set up in 10 μL binding buffer (100 mM HEPES(NaOH) pH 7.0, 150 mM NaCl, 5 mM $MgCl_2$, 2 mM dithiothreitol, 0.1 g/L bovine serum albumin, 0.2 g/L PolyC RNA, 10% (w/w) glycerol). The samples were incubated on ice for 30 min and products were separated by native gel electrophoresis on 6%

polyacrylamide Tris-glycine gels in Tris-glycine buffer on ice. Signals were visualized by phosphor screen autoradiography using a Typhoon scanner (GE Healthcare).

## Endonuclease assay

RNAs were chemically synthesized (Biomers) and labeled with $[\gamma]^{32}$P-ATP (Hartman Analytic) using T4 polynucleotide kinase (Thermo Fisher) or with the pCp-Cy5 fluorophore (Cytidine-5′-phosphate-3′-(6-aminohexyl)phosphate, Jena Bioscience) using the T4 RNA ligase enzyme (Thermo Scientific). RNA substrates were subsequently purified using Microspin G25 columns (GE Healthcare) or via gel electrophoresis on denaturing 7 M urea, 20% polyacrylamide Tris–borate–EDTA gels in 0.5-fold Tris–borate buffer. Reactions containing ~3 pmol $^{32}$P-labeled RNA and 5 pmol L protein or 0.015 pmol Cy5-labeled RNA and 2.5 pmol L protein were set up in 10 μL endonuclease buffer (100 mM HEPES(NaOH) pH 7.0, 150 mM NaCl, 50 mM KCl, 2 mM dithiothreitol, 0.1 mg/mL bovine serum albumin, 0.5 U/mL RNAsin (Promega)) and incubated at 37˚C for 1 h. The reaction was stopped via the addition of an equivalent volume of RNA loading dye (98% formamide, 18 mM EDTA, 0.025 mM SDS, xylene cyanole and bromophenol blue; xylene cyanole and bromophenol blue were omitted for fluorescence-based assays) and heating the samples at 95˚C for 3 min. Products were separated by gel electrophoresis on denaturing 7 M urea, 20% polyacrylamide Tris–borate–EDTA gels in 0.5-fold Tris–borate buffer. Signals were detected via phosphor screen autoradiography using a Typhoon scanner (GE Healthcare) for radioactivity-based assays. Fluorescent signals were detected in the gel using the Lourmat Fusion SL4 (Vilber) with an excitation wavelength of 624 nm and an emission filter for 695 nm wavelength.

## Polymerase assay

If not indicated otherwise, 6 pmol template 3′ RNA were incubated with 4 pmol L protein and NTPs (0.2 mM UTP/ATP/CTP and 0.1 mM GTP supplemented with 166 nM 5 μCi $[\alpha]^{32}$P-GTP) in 10 μL polymerase reaction buffer (100 mM HEPES(NaOH) pH 7.0, 150 mM NaCl, 50 mM KCl, 1 mM MnCl$_2$, 1% (w/w) glycerol, and 2 mM dithiothreitol). The samples were incubated at 30˚C for 1 h. The reaction was stopped via the addition of an equivalent volume of RNA loading dye (98% formamide, 18 mM EDTA, 0.025 mM SDS, xylene cyanole and bromophenol blue) and heating the samples at 95˚C for 3 min. Products were separated by gel electrophoresis on denaturing 7 M urea, 20% polyacrylamide Tris–borate–EDTA gels in 0.5-fold Tris–borate buffer. The Decade markers system (Ambion) was used as molecular weight marker. Signals were detected via phosphor screen autoradiography using a Typhoon scanner (GE Healthcare).

In samples containing 5′ RNA, the 5′ RNA was incubated with L protein for 15 minutes on ice to allow protein-RNA complex formation prior to the addition of template RNA and NTPs.

For primer elongation assays, 6 pmol template RNA were incubated with a tenfold excess of primers for 15 minutes on ice prior to the addition of L protein and NTPs to allow for primer annealing.

## Electron cryo-microscopy

A sample containing 0.2 mg/mL SNV L in EM buffer (30 mM HEPES pH 8.0, 250 mM NaCl, 2 mM dithiothreitol) was incubated with a tenfold excess of 5′ L 1–18 RNA on ice for 30 minutes. Glutaraldehyde (Thermo Scientific) was added to a final concentration of of 0.001% and incubated on ice for 15 minutes prior to the addition of 2 mM MS(PEG)8 (Thermo Scientific). 3.5 μL of sample were applied on UltrAuFoil 1.2/1.3 grids that had been glow-discharged at 25

mA for 45 s. Grids were blotted using a FEI Vitrobot Mark IV (blotting time: 3 s, blotting force 1, 100% humidity, 4˚C) before plunge-freezing in liquid ethane cooled to liquid nitrogen temperature. The grids were loaded into a 300-keV Titan Krios transmission electron cryo-microscope (Thermo Scientific) equipped with a K3 direct electron detector and a GIF BioQuantum energy filter (Gatan). A data collection of 4283 images was done using EPU at a nominal magnification of 105,000x with a pixel size of 0.85 Å and a defocus ranging from -0.8 to -2.2 μm. Each movie contains 50 frames and a total dose of 50 e⁻/Å² .

### Cryo-EM data processing and model building

Movies were realigned using Motioncor2 [41] and then imported in cryoSPARC 3.3.2 [42]. After CTF estimation, micrographs that showed an absence of ice or contamination by crystalline ice were manually removed (2337 micrographs were kept). The Blob picker utility with search diameter between 100 and 190 Å was used to pick 883,096 particles. Binned extracted particles were subjected to two rounds of 2D classification to clean the dataset. SNV L K124A particles were re-centered and re-extracted with a box size of 360x360 pixels$^2$ without binning (514,913 particles extracted). The CryoSPARC *ab initio* reconstruction utility was used to generate an initial model that was subsequently refined using the non-uniform refinement utility. Next steps were performed in Relion 4.0 [43,44]. Two rounds of 3D classification were performed with a 190 Å mask diameter and with global angular research for the first and with local angular research for the second. Particles belonging to classes showing densities for SNV L K124A core containing the 5′ L 1–18 RNA were merged (300,942 particles after the first 3D classification and 90,361 particles after the 2$^{nd}$ 3D classification). After the 2$^{nd}$ 3D classification, selected particles were used for a masked 3D refinement with local angular research. Finally, a post-processing step with a B-factor of -86.5 was done. For the final map, the global resolution is based on the FSC 0.143 cut-off criteria. Local resolution variation of this map was also estimated using Relion 4.0.

The obtained experimental map at 3.2 Å resolution was used for model building in Coot [45]. Specifically, the predictions of the regions spanning amino acids 726–898 and 899–1155 were docked into the map using the Dock-in-map function of PHENIX. The remaining regions were built *de novo* using AlphaFold2 predictions for guidance [34]. The model was refined in iterative cycles using Coot and PHENIX, and the final refinement was carried out setting the nonbonded weight to 500.0 [45,46].

## Supporting information

**S1 Table. Cryo-EM data collection, refinement and validation statistics.**
(DOCX)

**S1 Fig. Coomassie-stained SDS-PAGE of SNV L K124A after Strep-TactinXT purification.** SNV L K124A with a C-terminal His6-StrepII-tag was purified using Strep-TactinXT resin (see Materials and Methods) and eluted in several fractions. Shown are samples of the Lysate, the pelleted unsoluble fraction, the flow-through, the resin with bound residual protein, the first wash step as well as the elution fractions with a molecular weight marker (PageRuler Plus Prestained, Thermo Scientific). The shown elution fractions were pooled and further purified via heparin chromatography (HiTrap Heparin HP, GE Healthcare) to remove any bound RNA.
(TIFF)

**S2 Fig. RNA promoter binding to the SNV L protein. (A)** Electromobility shift assay to determine the binding capacity of the SNV L protein to the 5′ RNA 1–13 in different salt

concentrations. Radiolabeled 5′ RNA at a concentration of ~0.3 μM was incubated 0.2 μM L protein in EMSA buffer containing 500, 450, 400, 350, 300, 250, 200, 150, or 100 mM NaCl. The RNA-protein complexes were separated from free RNA by native PAGE and signals were visualized via phosphor screen autoradiography with a Typhon scanner (GE Healthcare). The control lane shows promoter RNA without L protein. **(B)** Electromobility shift assay to determine the recruitment of 3′ RNA 1–12 to the L protein in the presence of 5′ promoter RNA. Labeled 3′ RNA 1–12 at a concentration of ~0.3 μM was incubated with 0.2 μM L protein in the presence of unlabeled 5′ promoter RNA (nucleotides 1–13, 1–18, or 1–20) at an equimolar concentration or a two-fold excess of 5′ to 3′ RNA. Labeled 5′ RNA 1–18 was used as a positive control, a sample lacking L protein was used as a negative control.
(TIFF)

**S3 Fig. Promoter RNA secondary structure prediction.** The secondary structure of SNV L 5 1–18 RNA (5'-UAGUAGUAGACUCCGAGA-3') was carried out with RNAstructure [24].
(TIFF)

**S4 Fig. *In vitro* endonuclease actictivity with different RNA substrates.** A 10 μL reaction containing 0.5 μM full-length SNV L K124A, radiolabeled RNA substrate, and 0.1 mM (low MnCl2) or 10 mM (high) MnCl2 or MgCl2 in reaction buffer was incubated at 37˚C for one hour. The reactions were stopped by adding 10 μL of 2x RNA loading dye (98% formamide, 18 mM EDTA, 0.025 mM SDS, xylene cyanol, bromophenol blue) and heated to 95˚C for five minutes. RNA was separated on denaturing PAGE (25% acrylamide, 7 M urea, 0.5 x TBE) and visualized by autoradiography. The control lane shows a reaction without the addition of SNV L K124A.
(TIFF)

**S5 Fig. Template-dependence of the SNV L RdRp activity.** In a 10 μL reaction 4 pmol SNV L K124A was incubated with 6 pmol of the indicated RNA(s) **(A)** or increasing concentrations of SNV L 3′ 1-18P (6, 12, 20, or 40 pmol) **(B)**. NTPs supplemented with $[\alpha]^{32}$P-GTP were added, and the reaction was incubated at 30˚C for one hour. Reaction products were separated by denaturing PAGE and signals were visualized via phosphor screen autoradiography.
(TIFF)

**S6 Fig. Image processing strategy to obtain SNV L bound to 5′ 1–18 RNA cryo-EM map.** **(A)** Display of a representative micrograph after MotionCor2. The scale bar corresponds to 200 nm. **(B)** Representative 2D class averages. **(C)** 3D class averages of the 2$^{nd}$ 3D classification. The surrounded 3D class contains the particles selected for the last 3D reconstruction. The presence of the 5' 1–18 RNA forming the hook is indicated. The percentage of particles and the corresponding number is indicated below each class **(D)** Final reconstruction is displayed. Electron density map is colored according to the local resolution. Fourier Shell Correlation curves (FSC) and angular distribution of particles used in the final reconstruction are displayed.
(TIFF)

**S7 Fig. Density corresponding to RNA residues 2–12 of SNV L 5 1–18 RNA.** The RNA was built *de novo* in Coot [45] and was displayed as sticks in PyMol with the density displayed as mesh, contoured at 2.0 sigma within 1.5 Å of the displayed atoms.
(TIFF)

**S8 Fig. Predicted putative CBD of SNV L is less similar to LACV L CBD.** Structural comparison of the putative cap-binding domain within the C-terminal region of the hantavirus SNV L protein as predicted by AlphaFold2 [34] and the known structure of the phenuivirus

LACV L cap-binding domain (from PDB 7ORL). The side chains that may be involved in forming the aromatic sandwich to bind the cap are shown as orange sticks. The black dotted line corresponds to the distance between these two side chains as measured in PyMol. (TIFF)

**S9 Fig. Comparison of bunyavirus L protein CBD sequences.** The L protein CBD sequences of hantaviruses Sin Nombre virus (SNV, KT885044.1) and Hantaan virus (HTNV, ABD28179.1), phenuiviruses Rift Valley fever virus (RVFV, A2SZS3) and Severe fever with thrombocytopenia syndrome virus (SFTSV, I0DF35), and peribunyaviruses Bunyamwera virus (BUNV, A0A0A7KU93), and La crosse virus (LACV, A5HC98) were aligned using Clustal Omega [47]. Manual adjustments were made and the alignment was visualized with ESPript 3 [48] Shown is a comparison of the secondary structure elements of the predicted SNV L CBD and LACV L CBD (PDB 6Z6G). **(B)** The sequence identity and similarity between the different CBD within the alignment shown in **(A)** were analysed using the Sequence Identity And Similarity tool (http://imed.med.ucm.es/Tools/sias.html). (TIFF)

**S1 Alignment. Alignment of *Hantaviridae* and *Peribunyaviridae* full-length L protein sequences.** The L protein sequences of New World hantaviruses Sin Nombre virus (SNV, KT885044.1), Andes virus (ANDV, QRY27107.1), Bayou virus (BAYV, K7N869), and Black creek canal virus (BCCV, V5IVB1); Old World hantaviruses Hantaan virus (HTNV, ABD28179.1), Puumala virus (PUUV, ABN51178.1), Tula virus (TULV, A0A481S3H6), and Seoul virus (SEOV, A0A0B5JFL8), and peribunyaviruses Bunyamwera virus (BUNV, A0A0A7KU93), and La crosse virus (LACV, A5HC98) were aligned using Clustal Omega [47]. Manual adjustments were made and the alignment was visualized with ESPript 3 [48] Shown is a comparison of the secondary structure elements of SNV L and LACV L (PDB 6Z6G). (PDF)

## Acknowledgments

We want to thank Stephan Günther for his support throughout the entire project. We would also like to thank Harry Williams for helpful discussions and brainstorming interpretation sessions and Tomas Kouba and Benoît Arragain for discussions and feedback. We thank the cryo-EM Facility staff at the CSSB in Hamburg as well as the IBS in Grenoble and the Wilhelm und Maria Kirmser-Stiftung. IBS acknowledges integration into the Interdisciplinary Research Institute of Grenoble (IRIG, CEA). This work also used the platforms of the Grenoble Instruct-ERIC center (ISBG; UAR 3518 CNRS-CEA-UGA-EMBL) within the Grenoble Partnership for Structural Biology (PSB) supported by FRISBI (ANR-10-INBS-05-02). The ISBG; UAR 3518 was also supported by GRAL, financed within the University Grenoble Alpes graduate school (Ecoles Universitaires de Recherche) CBH-EUR-GS (ANR-17-EURE-0003). The electron microscope facility in Grenoble is supported by the Auvergne-Rhône-Alpes Region, the Fondation pour la Recherche Médicale (FRM), the fonds FEDER and the GIS-Infrastructures en Biologie Santé et Agronomie (IBiSA).

## Author Contributions

**Conceptualization:** Stephen Cusack, Kay Grünewald, Emmanuelle R. J. Quemin, Maria Rosenthal.

**Formal analysis:** Kristina Meier, Sigurdur R. Thorkelsson, Quentin Durieux Trouilleton, Dingquan Yu, Hélène Malet, Emmanuelle R. J. Quemin, Maria Rosenthal.

**Funding acquisition:** Stephen Cusack, Kay Grünewald, Maria Rosenthal.

**Investigation:** Kristina Meier, Sigurdur R. Thorkelsson, Quentin Durieux Trouilleton.

**Project administration:** Maria Rosenthal.

**Resources:** Kay Grünewald.

**Supervision:** Dominik Vogel, Jan Kosinski, Hélène Malet, Kay Grünewald, Maria Rosenthal.

**Validation:** Kristina Meier, Maria Rosenthal.

**Visualization:** Kristina Meier, Quentin Durieux Trouilleton.

**Writing – original draft:** Kristina Meier, Maria Rosenthal.

**Writing – review & editing:** Kristina Meier, Sigurdur R. Thorkelsson, Quentin Durieux Trouilleton, Dominik Vogel, Dingquan Yu, Jan Kosinski, Stephen Cusack, Hélène Malet, Kay Grünewald, Emmanuelle R. J. Quemin, Maria Rosenthal.

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
