## [Decision Letter · Decision Letter 0]

28 Mar 2023

Dear Dr. Rosenthal,

Thank you very much for submitting your manuscript "Structural and functional characterization of the Sin Nombre virus L protein" for consideration at PLOS Pathogens. As with all papers reviewed by the journal, your manuscript was reviewed by members of the editorial board and by several independent reviewers. In light of the reviews (below this email), we would like to invite the resubmission of a significantly-revised version that takes into account the reviewers' comments.

The reviewers and I appreciate the significance of this work. The reviewers made constructive comments most notably on the biochemistry as it relates to protein purity, and to activities. I agree with those comments. As addressing some of those comments may require additional experimentation the present recommendation is major revision.

We cannot make any decision about publication until we have seen the revised manuscript and your response to the reviewers' comments. Your revised manuscript is also likely to be sent to reviewers for further evaluation.

Sincerely,

Sean P.J. Whelan

Academic Editor

PLOS Pathogens

Matthias Schnell

Section Editor

PLOS Pathogens

Kasturi Haldar

Editor-in-Chief

PLOS Pathogens

orcid.org/0000-0001-5065-158X

Michael Malim

Editor-in-Chief

PLOS Pathogens

orcid.org/0000-0002-7699-2064

The reviewers and I appreciate the significance of this work. The reviewers made constructive comments most notably on the biochemistry as it relates to protein purity, and to activities. I agree with those comments. As addressing some of those comments may require additional experimentation the present recommendation is major revision.

Reviewer's Responses to Questions

**Part I - Summary**

Reviewer #1: In this manuscript, Meier and colleagues establish an expression and purification protocol for the L protein with a K124A endonuclease mutation of Sin Nombre virus (SNV). They perform functional characterisation of the L protein using RNA-binding, endonuclease and polymerase activity assays. In the second part of the manuscript, they present a 3D model of the L protein core region containing the RdRp domain, in complex with 5’ promoter RNA, obtained using single-particle cryo-EM. The authors compare their biochemical and structural observations with those for other segmented negative-sense RNA viruses (sNSVs).

Overall, this is an important addition to a growing body of literature on the structure and function of L proteins of sNSVs. The study highlights some interesting specificities of hantavirus L proteins such as the presence of the Hanta and California-like insertions in the palm domain of the RdRp. Generally, the manuscript is well-written and the data are clearly presented; for most parts the data support the conclusions but there are some issues that will need to be addressed.

Reviewer #2: This manuscript by Meier and coworkers examines the L protein polymerase of Sin Nombre hantavirus. Purification of the Sin Nombre virus L protein has proven problematic due to its highly active endonuclease activity, which suppresses protein expression. In this manuscript, the authors describe how they successfully purified the L protein by introducing a mutation into the endonuclease domain. The purified protein was analyzed in a series of biochemical assays that examined its endonuclease and RNA synthesis activities and the structure of the Sin Nombre virus L protein in association with RNA was determined. The manuscript adds to the growing body of literature regarding segmented negative strand RNA virus polymerases and provides valuable information that can be leveraged for future structure-informed functional studies. While the manuscript adds valuable new information to the field, some of the results from biochemical assays raised some questions that should be addressed.

**Part II – Major Issues: Key Experiments Required for Acceptance**

Reviewer #1: Fig. 2. I find it surprising that the radiolabelled RNA is almost fully degraded but there is no obvious depletion of the fluorescently labelled RNAs even at the highest metal concentrations. These data would be consistent with a phosphatase contamination in the L protein prep (could the authors include an SDS-PAGE of the L protein prep used?). The authors should check that the “endonuclease” activity they observe is sensitive to DPBA (as in the subsequent figure). The data in Fig. S3. (not referred to in the manuscript) suggest that the observed “endonuclease” activity is specific to polyA but the significance of this is not discussed. Given that all these data were obtained with an endonuclease mutant I wonder about the significance of these data; they seem to add little to an otherwise strong manuscript.

Fig. 3. In the in vitro polymerase activity assay the authors use an 18 long template but observe mostly longer and heterogeneous products that are unlikely to represent authentic transcription products. This issue is addressed in the Discussion and in fact the authors present a more convincing result using a 3’ phosphorylated RNA template (Fig. S4) where a major 18 nt product is observed with some longer products that can be explained by template switching. I wonder why the authors don’t show this result in the Results section instead of data in Fig. 3 that are difficult to interpret. A further issue with the data in Fig. 3B is that the DPBA experiment lacks a positive control; also, DPBA seems to change the band pattern of products in the Mg2+ conditions but this is not commented on. Comparing band patterns in panel A and B I wonder whether the label for Mn2+ and Mg2+ is switched around. Panel A lacks size markers.

Reviewer #2: 1. It would be helpful to show the purified polymerase on a Coomassie stained denaturing polyacrylamide gel to provide an indication of polymerase purity.

2. Fig. 2: Panel A does not appear to be consistent with panel B, lanes 1-5. In panel A, 40 mer poly A was efficiently cleaved by the endonuclease, whereas in B a 27 mer poly A showed minimal cleavage. It should be determined if this is due to the RNA length or the presence of the fluorescent tags.

3. It is possible that the degradation observed in Fig. 2 is due to a contaminating RNAse given that the activity is not ablated by the K124A substitution. Are there additional mutations that could be introduced to ascertain if the endonuclease activity that is observed is due to the SNV L protein? Alternatively, if DBPA has specificity for viral endonucleases perhaps this could be used to confirm the specificity of this cleavage.

4. Fig. 3: It appears that the data presented in Fig. 3A and Fig. 3B are inconsistent with each other. In panel A, 10 mM MnCl2 appears to completely inhibit RNA synthesis activity, whereas this is not the case in Fig. 3B. If this is because the panel B was exposed for much longer than panel A, this should be stated in the results section or legend. If there is another reason, an explanation should be provided.

5. Fig. 3B: The pattern of RNA product bands is very different in the reactions that contained MgCl2 +/- DBPA. An explanation should be provided for these results.

6. Fig. 3 and subsequent RNA gels, there are products as long as 40 nt in length, although the template is only 18 nt. In the discussion section (lines 518-522) it is suggested that this is due to template switching. However, in Fig. 5C, there were bands of ~15 and 18 nt, even in reactions lacking GTP. It is difficult to conceive how the polymerase generates products of 15 and 18 nt in length by template switching in -GTP conditions. Is it possible that the RNA is not fully denatured? In addition, the discrepancy between the template and product length should be mentioned in the Results section.

**Part III – Minor Issues: Editorial and Data Presentation Modifications**

Reviewer #1: Fig. 1 and line 143. These data are suggestive that the interaction of the L protein with the 5’ RNA is stronger than with the 3’ RNA but no affinity measurements were performed to be able to conclude that the interaction is “significantly stronger”. Please tone down the language.

Fig. 4. Size markers are lacking.

Line 322. Fig. S1A is referred to here; is this correct?

Fig. 5. The authors suggest that there is a single binding site for the di/tri nucleotides (template sequence at the bottom of panel A). Due to the repetition of the AUC motif at the 3’ end these primers can all bind at multiple sites in the template; should this be considered?

Fig. 8 and Fig. S7. These two figures could be joined as a single supplementary figure and perhaps discussed in the Discussion rather than being presented as Results.

Fig. S1A. The legend refers to 3’ RNA but 5’ RNA is shown in the figure.

Fig. S5C. The authors indicate only one class contains RNA. It would be informative to include particle numbers for these classes to give an idea of RNA occupancy.

Fig. S6. Include nt labels to orient the reader.

Table S1 and line 668. The table states about 500k particles were used in the final reconstruction; however in the methods 90,361 are mentioned. Please check.

Line 528. “extent”

Line 589. Presumably “300 mM NaCl”.

Lines 595 and 598. No “reaction” occurs in an electrophoretic mobility shift assay.

Line 625. Please check that 0.5 mM GTP is correct; this would result in a very low ratio of 32P-GTP most likely not sufficient for detecting the RNAs.

Reviewer #2: 7. Line 299-300 – “ Taken together, our data suggest that the SNV L RdRp has a higher specificity for purines, indicative of a low RdRp fidelity”. This is worded oddly. It would make more sense if it was written as “Taken together, our data suggest that the SNV L RdRp has a low specificity for pyrimidines, indicative of a low RdRp fidelity”.

8. Line 475: the inhibitory effect of the 5´ RNA was reduced in the presence of short primers. Could this be because the primer competed with 5´ RNA for binding to the 3´ RNA? Is the sequence of SNV different than for other viruses, do the 5´ and 3´ RNAs have a higher Tm?

PLOS authors have the option to publish the peer review history of their article (what does this mean?). If published, this will include your full peer review and any attached files.

Reviewer #1: No

Reviewer #2: No
---

## [Decision Letter · Decision Letter 1]

27 Jun 2023

Dear Dr. Rosenthal,

Thank you very much for submitting your manuscript "Structural and functional characterization of the Sin Nombre virus L protein" for consideration at PLOS Pathogens. As with all papers reviewed by the journal, your manuscript was reviewed by members of the editorial board and by several independent reviewers. The reviewers appreciated the attention to an important topic. Based on the reviews, we are likely to accept this manuscript for publication, providing that you modify the manuscript according to the review recommendations.

The manuscript is in principle accepted but please respond to the minor comment of reviewer 2.

Sincerely,

Sean P.J. Whelan

Academic Editor

PLOS Pathogens

Matthias Schnell

Section Editor

PLOS Pathogens

Kasturi Haldar

Editor-in-Chief

PLOS Pathogens

orcid.org/0000-0001-5065-158X

Michael Malim

Editor-in-Chief

PLOS Pathogens

orcid.org/0000-0002-7699-2064

The manuscript is in principle accepted but please respond to the minor comment of reviewer 2.

Reviewer Comments (if any, and for reference):

Reviewer's Responses to Questions

**Part I - Summary**

Reviewer #1: The authors have revised their paper satisfactorily addressing all the reviewers' comments. I have no further comments.

Reviewer #2: The reviewers' comments have been addressed and the conclusions appear to be sound. The only query I have is that on lines 210-213 it is stated that "we also detected degradation of the Cy5-labeled viral 3' and 5' RNAs 1-18 in substoichiometric RNA concentrations relative to the L protein, albeit with lower efficiency than the poly A substrate". However, looking at panel 2B, it is difficult discern a difference in cleavage of the polyA RNA versus the 3' 1-18 or 5' 1-18 RNAs. Revision of the accompanying text or an explanation would be helpful.

**Part II – Major Issues: Key Experiments Required for Acceptance**

Reviewer #1: (No Response)

Reviewer #2: (No Response)

**Part III – Minor Issues: Editorial and Data Presentation Modifications**

Reviewer #1: (No Response)

Reviewer #2: (No Response)

PLOS authors have the option to publish the peer review history of their article (what does this mean?). If published, this will include your full peer review and any attached files.

Reviewer #1: No

Reviewer #2: No

Figure Files:

Data Requirements:

Reproducibility:

References:

---

## [Editor Report · Decision Letter 2]

4 Jul 2023

Dear Dr. Rosenthal,

We are pleased to inform you that your manuscript 'Structural and functional characterization of the Sin Nombre virus L protein' has been provisionally accepted for publication in PLOS Pathogens.

Best regards,

Sean P.J. Whelan

Academic Editor

PLOS Pathogens

Matthias Schnell

Section Editor

PLOS Pathogens

Kasturi Haldar

Editor-in-Chief

PLOS Pathogens

orcid.org/0000-0001-5065-158X

Michael Malim

Editor-in-Chief

PLOS Pathogens

orcid.org/0000-0002-7699-2064
---

## [Editor Report · Acceptance letter]

21 Jul 2023

Dear Dr. Rosenthal,

We are delighted to inform you that your manuscript, "Structural and functional characterization of the Sin Nombre virus L protein," has been formally accepted for publication in PLOS Pathogens.

Best regards,

Kasturi Haldar

Editor-in-Chief

PLOS Pathogens

orcid.org/0000-0001-5065-158X

Michael Malim

Editor-in-Chief

PLOS Pathogens

orcid.org/0000-0002-7699-2064